# Cryo-EM structure of *Chlamydomonas reinhardtii* Photosystem I complexed with cytochrome $c_6$

Yu Ogawa [1,6], Gyana Prakash Mahapatra[2,6], Yuval Milrad [1,6], Michelle Schimpf[1], Genji Kurisu [3], Michael Hippler [1,4] ✉ & Jan Michael Schuller [2,5] ✉

Photosynthetic electron transfer relies on small soluble carriers that shuttle electrons between the cytochrome $b_6f$ complex and Photosystem I (PSI). While copper-containing plastocyanin (Pc) serves this role in plants, the heme protein cytochrome $c_6$ (Cyt $c_6$) is also employed in algae and cyanobacteria. Here, we present a cryo–electron microscopy structure of a Cyt $c_6$:PSI complex from *Chlamydomonas reinhardtii*. We observe that the heme group of Cyt $c_6$ is positioned ~11 Å away from P700, stabilized by extensive contacts involving a N-terminal helix-loop-helix motif of PSAF, characteristic of eukaryotic PSI. Notably, the algal Cyt $c_6$ also retains an arginine residue (R66) which is crucial for cyanobacterial donor:PSI reactions. Our structure reveals the previously uncharacterized interactions involving this residue; it can form a putative electrostatic contact with PsaB-D623 while also contributing to a tri-planar π(cation)-π interactions with adjacent residues. Our findings provide a structural framework for understanding the mechanism and evolution of donor:PSI interactions.

The photosynthetic electron transport chain proceeds through a series of finely tuned bimolecular electron transfer reactions essential for maintaining efficient photosynthesis. This chain begins with Photosystem II (PSII), which catalyzes the oxidation of water and the reduction of plastoquinone (PQ). The reduced plastoquinol (PQH₂) is then oxidized by the Cytochrome $b_6f$ complex (Cyt $b_6f$), transferring electrons to water-soluble carriers such as cytochrome $c_6$ (Cyt $c_6$) or plastocyanin (Pc), which in turn donate electrons to Photosystem I (PSI). This electron transfer through Cyt $b_6f$ is coupled to proton translocation across the thylakoid membrane, generating an electrochemical proton gradient that drives the synthesis of adenosine triphosphate (ATP).

From an evolutionary standpoint, the ancestral PSI is thought to have interacted with the heme-based electron carrier Cyt $c_6$[1,2].

However, over time, due to selective pressures such as fluctuations in metal bioavailability, the copper-containing protein Pc emerged as a functional replacement for Cyt $c_6$. In contemporary photosynthetic organisms, this transition is reflected in the diversity of electron carriers; certain cyanobacteria still use only Cyt $c_6$, other cyanobacterial and algal species switch between Cyt $c_6$ and Pc depending on iron/copper availability, while vascular plants rely exclusively on Pc[3]. Despite differences in metal cofactors, Cyt $c_6$ and Pc have undergone convergent evolution and exhibit similar reaction kinetics with their partners, although these kinetics vary significantly among species[1,2,4,5]. In particular, the kinetics of PSI reduction by Cyt $c_6$/Pc have been extensively studied using laser-flash absorption spectroscopy. In green algae and vascular plants, the kinetics typically show a fast first-order

[1]Institution of Plant Biology and Biotechnology, University of Müenster, Schlossplatz 8, Münster, Germany. [2]Philipps-University Marburg, Department of Chemistry and Center for Synthetic Microbiology (SYNMIKRO), Karl-von-Frisch-Strasse 14, Marburg, Germany. [3]Institute for Protein Research, Osaka University, 3-2 Yamadaoka, Suita, Osaka, Japan. [4]Institute of Plant Science and Resources, Okayama University, 2-20-1 Chuo, Kurashiki, Okayama, Japan. [5]Microbes-for-Climate (M4C) Cluster of Excellence, Marburg, Germany. [6]These authors contributed equally: Yu Ogawa, Gyana Prakash Mahapatra, Yuval Milrad. ✉e-mail: mhippler@uni-muenster.de; jan.schuller@synmikro.uni-marburg.de

intramolecular electron transfer followed by the slower second-order bimolecular reaction[2,4–6]. The former suggests stable donor:PSI complex formation, while the latter reflects the interaction of freely diffusing donors and PSI. In contrast, in many cyanobacteria, the slower biomolecular phase is dominant or exclusive, particularly in vitro[2,4,5].

From a structural perspective, deciphering such donor:PSI interactions proved to be challenging. Insights from early biophysical studies, supported by mutagenesis experiments, indicated that the formation of these complexes is governed by the so-called northern hydrophobic patch on the electron carriers (Cyt $c_6$/Pc) docking into a hydrophobic groove at the PsaA–PsaB interface[7–14]. Moreover, in the case of eukaryotic systems, the interaction is also driven by the negatively charged residues on the east patch of donors, which engage in salt bridges with Lys residues in the extended N-terminal region of PSAF[9,11,12,15–20]. Recently, these estimation has been validated by the cryo-EM (Cryogenic electron microscopy) analysis of eukaryotic Pc:PSI complexes from both *Pisum sativum* and *Chlamydomonas reinhardtii*[21–23]. However, to date, there is no such structural evidence showing exactly how Cyt $c_6$ binds to PSI, as most recent attempts (mainly on cyanobacterial systems) have yet to meet a resolution that would be sufficient for drawing biophysical conclusions[24,25]. This gap in structural knowledge still limits our understanding of the evolution and mechanistic diversity of donor:PSI reactions across photosynthetic lineages.

To address this gap, we perform cryo-EM analysis of a *C. reinhardtii* Cyt $c_6$:PSI complex, stabilized via chemical crosslinking that preserves a conformation competent for efficient electron transfer[18,26]. This work reveals that Cyt $c_6$ utilizes its negatively charged residues to interact with PSAF, similarly to eukaryotic Pc, while it has an Arg residue (R66) that also significantly contributes to binding and electron transfer to PSI. According to sequence data on various photosynthetic branches, this Arg seems to be absent in eukaryotic Pc, while it is well conserved and functionally essential in cyanobacterial systems (Supplementary Fig. 1)[13,27,28]. Therefore, our structure offers a unique structural window into ancestral donor:PSI interactions, providing a structurally tractable model of their evolution.

## Results

### Chlamydomonas Cyt c₆: PSI complex

To obtain structural insights into the Cyt $c_6$:PSI interactions, we reconstituted a Chlamydomonas Cyt $c_6$:PSI complex by chemical crosslinking (Supplementary Fig. 2). PSI (PsaB-His$_{20}$) was affinity-purified from Chlamydomonas cells grown under standard conditions, while Chlamydomonas Cyt $c_6$ was recombinantly expressed, purified, and pre-activated with 1-ethyl-3-[3- dimethylaminopropyl] carbodiimide (EDC) and *N*-hydroxysulfosuccinimide (sulfo-NHS)[23]. This converts the Cyt $c_6$ carboxyl groups into a short-lived sulfo-NHS ester intermediate (Supplementary Fig. 2a). After removing excess crosslinking agents, the activated Cyt $c_6$ was incubated with the isolated PSI complex (Supplementary Fig. 2b, c). As Cyt $c_6$:PSI complexes form, sulfo-NHS esters located close to PSI amino groups can react to produce amide bonds, enabling zero-length crosslinking, while the esters that are not in close vicinity are rapidly hydrolyzed without forming crosslinks (Supplementary Fig. 2c)[17,18,26,29,30]. This stabilizes the donor:PSI complexes at the positions where carboxyl and amino groups are in close proximity. Importantly, this type of cross-linking preserves the native Cyt $c_6$:PSI complex conformation that is competent in rapid electron transfer[18,26]. The cross-linked Cyt $c_6$:PSI complex was subjected to sucrose density gradient (SDG) centrifugation to separate it from the non-cross-linked Cyt $c_6$ (Fig. 1a). Successful crosslinking was confirmed by the appearance of a ~28.5 kDa band corresponding to the Cyt $c_6$:PSAF product and by the concurrent depletion of the free PSAF band (~20 kDa) (Fig. 1b)[18]. Thereafter, the cross-linked complex was subsequently subjected to cryo-EM single particle analysis (SPA) for structural characterization (Supplementary Fig. 3).

Cryo-EM SPA of the sample yielded a reconstructed map with a global resolution of 1.83 Å, in which PSI was resolved with near-atomic detail (Fig. 1c, d, Supplementary Figs. 3–5 and Supplementary Table 1). The structure closely resembled previously reported Chlamydomonas PSI monomers, and the map quality allowed for precise modeling of all core subunits and cofactors (Fig. 1c, d, Supplementary Fig. 5)[23,31–34]. The PSI model comprises 19 protein subunits and includes 189 chlorophyll a, 28 chlorophyll b, 27 β-carotenes, 2 neoxanthins, 7 xanthophylls, 16 luteins, 2 phylloquinones, 3 [4Fe–4S] clusters, and 63 lipid molecules, along with 1019 water molecules. Although weak density was initially observed near PSAF in the global map, consistent with Cyt $c_6$, the signal was insufficient for confident interpretation (Supplementary Fig. 3). To improve visualization of Cyt $c_6$, we performed focused refinement in CryoSPARC using a soft mask around the expected binding site[35,36]. This yielded a locally enhanced reconstruction at 2.06 Å, in which an additional density adjacent to PSAF was clearly resolved (Supplementary Fig. 4). The shape and volume of the density matched that of Cyt $c_6$, enabling us to rebuild the Cyt $c_6$ model into the density, based on the Chlamydomonas Cyt $c_6$ crystal structure (PDB:1CYJ)[37]. Alongside the Cyt $c_6$-bound PSI particles, SPA analysis identified a subpopulation of PSI particles lacking bound Cyt $c_6$, which was reconstructed separately (Supplementary Fig. 6). Structural comparison of the Cyt $c_6$-free PSI reconstruction with the Cyt $c_6$-bound PSI revealed no detectable structural differences. Superposition of the two models yielded a root-mean-square deviation (RMSD) of 0.189 Å, indicating that Cyt $c_6$ binding does not induce large-scale structural rearrangements in PSI.

In addition to the luminal-side Cyt $c_6$ density, we also observed two non-protein assigned densities on the stromal side of the PSI complex, located near the PSAD and PSAE subunits (Supplementary Fig. 3c). These densities were consistently present across multiple reconstructions but remained insufficiently resolved for interpretation, likely due to extreme conformational flexibility or heterogeneous occupancy. Such densities were not detected in our previous analysis, where the PSI fractions migrated to upper densities compared with the current work, suggesting the PSI complexes contain other components (Fig. 1a)[23]. The positions of the stromal densities are reminiscent of those previously reported in PSI structures from Chlamydomonas, where they were proposed to correspond to transient or regulatory interaction partners such as FNR or loosely bound ferredoxin[38]. While the exact identity of the densities in our structure remains unclear, their consistent localization suggests a potential functional relevance.

### Cyt c₆:PSI binding

The donor binding site of PSI accommodates a shallow hydrophobic pocket, formed at the luminal interfaces of PsaA and PsaB subunits (Supplementary Fig. 7a, gray circle). This pocket is delineated at its base by the symmetrical *l/l'* loops (Supplementary Fig. 7b, light green and dark green rectangles, respectively), which hold a Trp dimer in its center (PsaA-W651 and PsaB-W626, Fig. 2a, Supplementary Fig. 7b). This region is partly enclosed by the positively-charged protrusion of the N-terminal domain of PSAF. It is important to note that the hydrophobic PsaA-PsaB region is highly conserved throughout the full range of photosynthetic organisms, whereas the positively charged helix-loop-helix motif of PSAF is only present in eukaryotic organisms (Supplementary Fig. 1)[39].

Our data revealed that the binding of Cyt $c_6$ to PSI is stabilized by an extensive network of non-covalent interactions. They include the electrostatic interactions between the positively charged residues on PSAF and the negatively charged residues on Cyt $c_6$. In particular, PSAF-K23 and K27 were in the immediate vicinity of Cyt $c_6$-E69, and between the PSAF-K27 and Cyt $c_6$-E69, our high-resolution density map observed a well-defined density corresponding to the covalent crosslinking (Fig. 2a). Here, we count PSAF residues starting from the N-terminus of the mature protein, eliminating the transit peptide

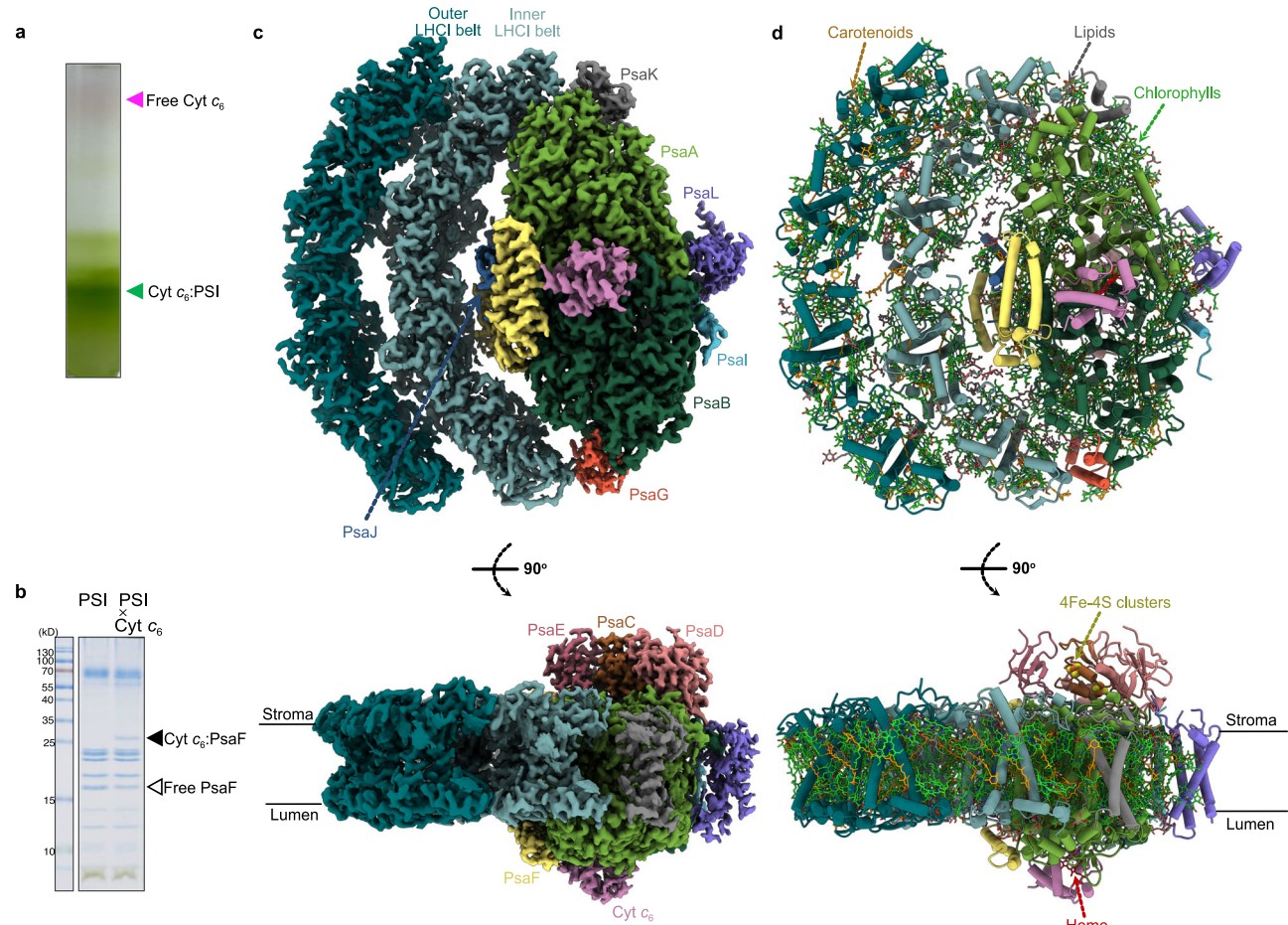

**Fig. 1 | Purification and overall structure of Cyt $c_6$:PSI complex. a** Purification of cross-linked Cyt $c_6$:PSI particles by sucrose density gradient centrifugation. **b** SDS-PAGE of PSI and cross-linked Cyt $c_6$:PSI particles. The same experiments were performed twice, and a representative result is shown. Source data are provided as a Source Data file. **c** Electron density map of Cyt $c_6$:PSI complex (top: luminal view; bottom: side view). **d** Structural model of Cyt $c_6$:PSI complex (top: luminal view; bottom: side view).

segment, for consistency with earlier reports. However, this segment is included in the PDB data and the LigPlot⁺ analysis mentioned below. The other charged residues on PSAF and Cyt $c_6$ were not located in sufficient proximity to enable salt bridge formation ( < 6Å) (Fig. 2b)[40,41]. However, this observation should be addressed with caution, as the atomic distances shown here are approximate values derived from a well-resolved atomic model.

The involvement of other specific residues and water molecules was identified using LigPlot⁺ software (Fig. 2a, Supplementary Figs. 8, 9a)[42,43]. Within the shallow pocket, on the PsaA-$l$ loop, a water-mediated hydrogen bond (HOH196) connects the PsaA-Q656 and Cyt $c_6$-G58, while PsaA-Q659 hydrophobically interacts with Cyt $c_6$-P61/A62 (Supplementary Fig. 8a and 9b, left). The PsaB-$l$ loop provides more extensive contacts among non-charged residues and the heme group of Cyt $c_6$ (Supplementary Figs. 8b, d, 9b, right). One interesting exception is the case of a conserved charged pair of PsaB-R622/D623, which interacts with Cyt $c_6$-R66. Although PsaA-$l$ loop also contains the structurally symmetric Arg/Asp pair (R647/D648), it is not directly engaged in Cyt $c_6$ binding. The interface over the shallow pocket is further reinforced by additional peripheral contacts; specifically, PsaA-G662 interacts with Cyt $c_6$-D65, PsaB-Q604/N606 with Cyt $c_6$-N11/G12, and PsaB-R733 with Cyt $c_6$-A16 (Supplementary Figs. 8a, b, 9c). The Cyt $c_6$-D65 also forms a hydrogen bond with PSAF-K20, extending the peripheral contact site on PsaA into PSAF (Supplementary Figs. 8c, 9c, top).

As described above, the R622/D623 pair on the PsaB-$l'$ loop engages Cyt $c_6$-R66; the corresponding Arg residue is known to be critical for PSI binding and electron transfer in cyanobacteria (Fig. 2c, d, Supplementary Fig. 8b)[27,28]. To better understand the functional importance of this residue, we examined its local environment beyond the direct PsaB interface (Fig. 2d). Cyt $c_6$-R66 may form an electrostatic interaction with PsaB-D623, while it also contacts PsaB-R622, possibly causing electrostatic repulsion. Rather, it is also possible that these two adjacent PsaB residues partially neutralize each other. Additionally, the hydroxyl groups of PsaB-S613 and PsaB-Y616 as well as a water molecule (HOH116) can further delocalize the local charges. This microenvironment suggests that electrostatics may not fully account for the essential role of the corresponding Arg residue in cyanobacteria. Notably, within the Cyt $c_6$ protein, R66 sits directly above W63 and is postulated to participate in π(cation)-π interactions[44,45]. Here, we found that in the Cyt $c_6$:PSI complex, the indole group of W63 and the guanidinium groups of R66 and PsaB-R622 are arranged in a semi-parallel stack at distances of 3–5 Å. This suggests a potential three-layer π(cation)-π interaction that could further stabilize the donor:acceptor interface. Although PsaB-R622 was previously predicted to electrostatically interact with Cyt $c_6$-D65[13], this Cyt $c_6$ residue instead seems to make the peripheral contacts on PsaA and PSAF as mentioned above (Supplementary Figs. 8a, c, 9c, top).

Together, the Cyt $c_6$:PSI binding is supported by a broad, multivalent interaction surface involving polar, charged, hydrophobic and solvent-mediated elements, as well as π(cation)-π interactions. In the complex, the redox cofactors (i.e., the heme of Cyt $c_6$ and the P700 chlorophyll pair of PSI) are positioned at

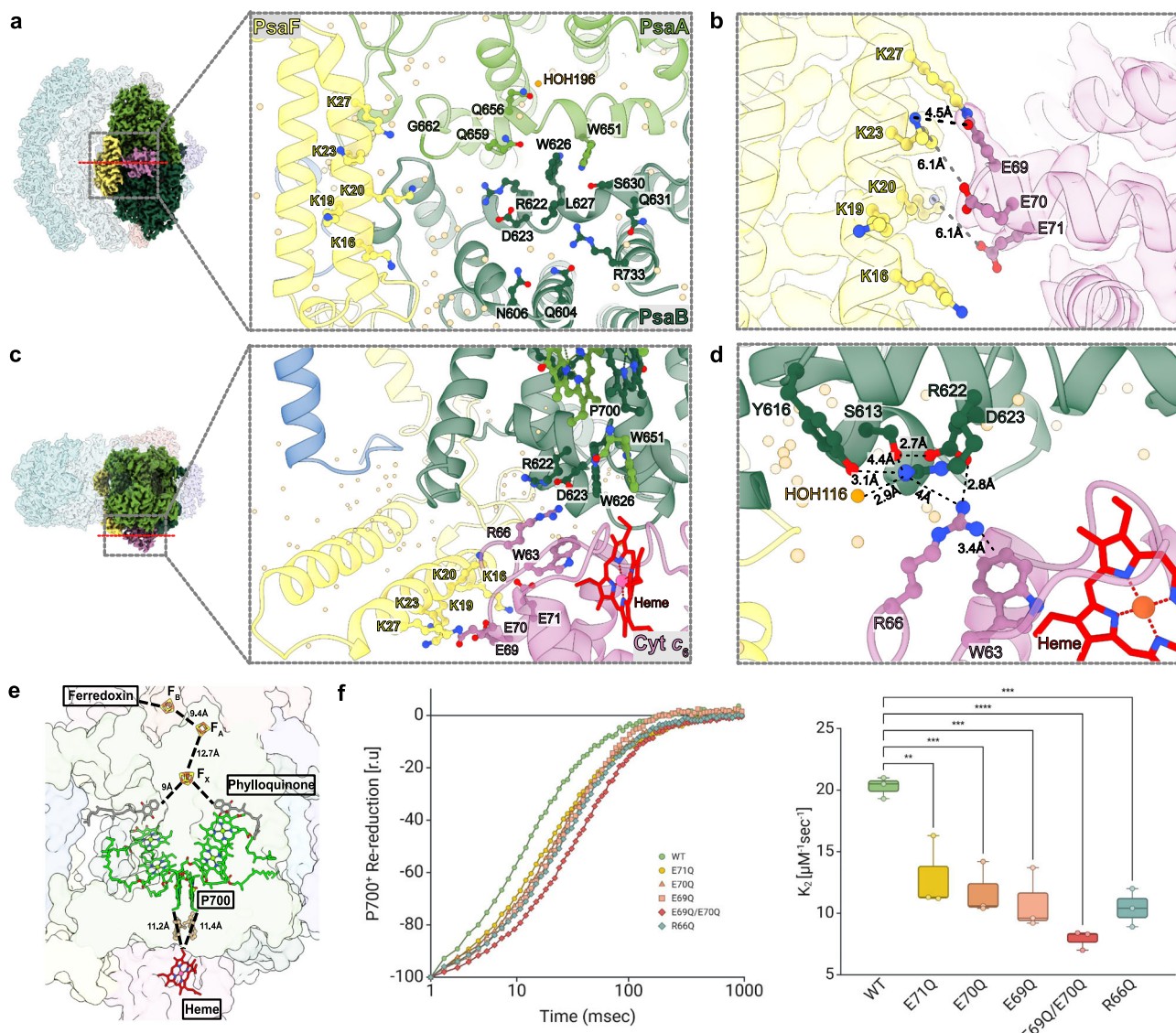

**Fig. 2 | Cyt $c_6$:PSI binding. a** Luminal view of Cyt $c_6$:PSI interface (just PSI side on the right panel). Cyt $c_6$, PsaA, PsaB and PsaF are colored in pink, light green, dark green and yellow, respectively. Directly interacting amino acids and water molecules (orange) are shown, in addition to PsaA-W651 and PsaF-K16/K19/K20. **b** Cyt $c_6$:PsaF interface with the corresponding cryo-EM density. The putatively involved residues are shown with their distances. **c** Side view of Cyt $c_6$:PSI interface. P700, heme and the Trp dimer are shown, as well as the amino acid residues putatively involved in PsaF-dependent electrostatic interactions and Cyt $c_6$-R66-mediated protein-protein interactions. **d** Cyt $c_6$:PsaB interface. The amino acid residues and a water molecule putatively involved in the protein-protein interactions mediated by R66 are shown with their distances. **e** Edge-to-edge distances of electron-transfer molecules in Cyt $c_6$:PSI complex. **f** in vitro kinetic analysis of P700$^+$ re-reduction using Cyt $c_6$ variants with mutations at putatively interacting amino acid residues. Box-plots (right) (center line, median; box limits, upper and lower quartiles; whiskers, 5 to 95 percentile; points outliers) show the averaged K$_2$ values of 2–4 μM Cyt $c_6$ (3 biological repetitions). Statistical analysis was conducted using a One-way ANOVA with Dunnett multiple comparisons test (**, *** and **** indicate $P$-values smaller than 0.01, 0.001 and 0.0001, respectively). The illustration of statistical analysis was created in BioRender. Milrad, Y. (2026) https://BioRender.com/egh38am. Source data are provided as a Source Data file.

an edge-to-edge distance of 11.2–11.4 Å, placing them in a suitable range for efficient electron transfer (Fig. 2e). We also observed that these two cofactors encompass the Trp dimer situated in the center of the l/l′ loops. Moreover, the tight contact on this region excludes surrounding water molecules from the redox interface, presumably lowering the energetic barrier for electron transfer (Fig. 2a, Supplementary Fig. 7b).

### Significance of eukaryote/cyanobacteria-like interactions

Green algal Cyt $c_6$ is uniquely positioned in evolution, utilizing PSAF-mediated electrostatic interactions just as eukaryotic Pc does, while conserving R66, a residue known to be essential for PSI interaction in cyanobacteria (Supplementary Fig. 1)[13,26]. To test the roles of these

interactions, we introduced several single mutations into Cyt $c_6$, which eliminate its negative charges (E69Q, E70Q, E71Q), in addition to a double mutant (E69Q/E70Q). Moreover, to verify the role of R66, we generated another mutant variant, namely R66Q. To evaluate the impact of these mutations, we measured P700$^+$ re-reduction kinetics and assessed second-order rate constants (k$_2$) (Fig. 2f)[39]. Our results show that all variants exhibit a significant decrease in k$_2$ compared to wild-type Cyt $c_6$, with the most severe impairment observed for the E69Q/E70Q double mutant. The E71Q variant showed a more modest reduction, suggesting a less critical role for this residue.

We then assessed the impact of these mutations on complex stability using chemical cross-linking (Supplementary Fig. 10). Accordingly, R66Q, E69Q, and E70Q mutations drastically affected the

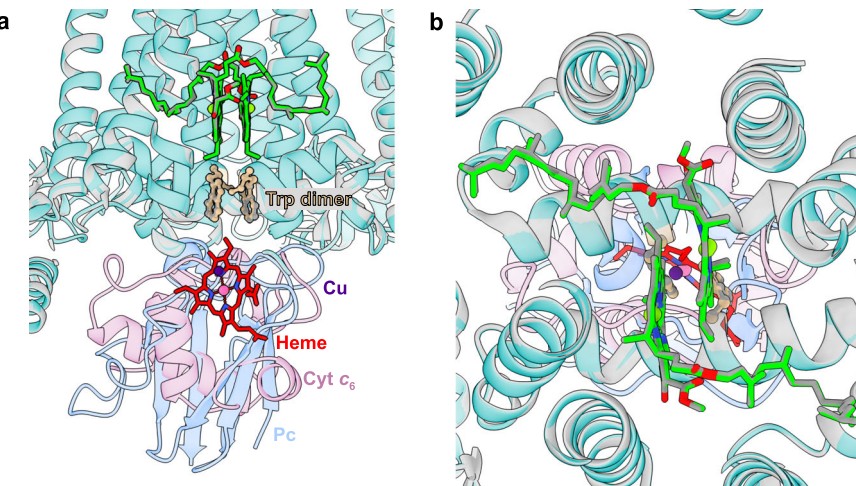

**Fig. 3 | Comparative overlay of donor:PSI complexes. a** Overlay of the Cyt $c_6$:PSI on the *Chlamydomonas* Pc:PSI (PDB ID 7ZQE, 7ZQC), in accordance with spinach Pc:PSI (PDB ID 6ZOO). The former PSI is colored in light blue, the latter in gray.

P700, the Trp dimer and the cofactor of each donor are shown. **b** Stromal view of the overlayed structures of *Chlamydomonas* Cyt $c_6$:PSI and Pc:PSI.

cross-link formation between Cyt $c_6$ and PSAF, supporting their direct involvement in intermolecular interactions. In contrast, E71Q showed little effect on cross-linking efficiency, consistent with its weaker kinetic phenotype. Interestingly, the phenotype of the E69Q/E70Q double mutant was weaker than that of every single mutant, suggesting the formation of donor:acceptor complexes in different conformational states, which are less competent for the electron transfer reactions.

Overall, these results demonstrate that the interactions mediated by the conserved negative residues (in particular, E69 and E70) and R66 of Cyt $c_6$ are both critical for stable PSI binding and efficient electron transfer. These observations underscore green algal Cyt $c_6$ as a unique evolutionary intermediate of PSI donors, which harnesses the ancestral features.

## Discussion

In this work, we present a high resolution cryo-EM structures of the Chlamydomonas PSI core at 1.83 Å and its Cyt $c_6$-bound complex at 2.06 Å, revealing detailed molecular interactions where the ancestral features coexist. This high resolution enables us to draw conclusions about donor:PSI complex formation. Of note, the conformation of Cyt $c_6$ binding reported here is dissimilar to those predicted by modeling in previous studies[46–48]. In our structure, we observed that the interface is extensively stabilized by a network of hydrogen bonds, salt bridges, hydrophobic contacts and π(cation)-π interactions primarily involving the $l$/$l'$ loops and the N-terminal domain of PsaF (Fig. 2, Supplementary Figs. 7–9). The comprehensive disclosure of the interactions within the shallow pocket indicates asymmetric binding on PsaA and PsaB, with PsaB contributing a larger number of interacting residues (Fig. 2a, Supplementary Figs. 8, 9). Cyt $c_6$ adheres more closely to PsaB and contacts with PsaA at limited sites, while positioning the heme right beneath the Trp dimer and P700 (Fig. 2e). Relying on fewer contact sites in interacting with PsaA may explain the stronger functional impacts of mutations on PsaA-W651 relative to PsaB-W626[10,13]. This interaction network on PSI core ensures the stringent donor:acceptor orientation; the Cyt $c_6$ heme is positioned approximately 11 Å from the P700 chlorophyll dimer, with the conserved Trp residues located along the electron transfer axis. This architecture is consistent with reports of rapid electron transfer kinetics (~4 µs), a characteristic of native as well as of cross-linked Cyt $c_6$:PSI complex[18,26].

Superimposition of Cyt $c_6$:PSI on the Pc:PSI from Chlamydomonas (PDB: 7ZQE, 7ZQC, in accordance with the structure of Pc:PSI from

Spinach, PDB: 6ZOO) enabled us to compare the electron transfer pathways created by the alternative donors (Fig. 3)[21,23]. First, we observed the co-localization of the heme iron of Cyt $c_6$ and the copper of Pc, positioned directly below the Trp gateway and P700. In addition, we noted similar exclusion of surrounding water molecules from the interface, which couples electron transfer to minimal solvent reorganization (Fig. 2a, Supplementary Fig. 7b)[22,23]. Taken together, these observations suggest that transferred electrons might be relayed via a π-conjugated indole system en route to the reaction center, highlighting the functional equivalence of these two evolutionarily distinct donors.

The structure also allows dissecting the electrostatic interactions mediated by the N-terminal domain of PSAF, characteristic of vascular plants and green algae. The covalent cross-link between PSAF-K27 and Cyt $c_6$-E69, resolved by cryo-EM captures a dominant and highly uniform binding conformation, validating the complex structure as a faithful representation of the physiological electron transfer assembly (Fig. 2b). The mutagenesis analysis revealed that E69 and E70 are critical for complex formation and electron transfer, while E71 plays a more limited role (Fig. 2f, Supplementary Fig. 10). Similarly, previous mutagenesis studies showed that the positively charged residues on the N-terminal domain of PSAF are necessary for complex stability and efficient electron transfer[19]. However, despite their important roles, charged residues other than PSAF-K23/K27 and Cyt $c_6$-E69 were too far apart to form salt bridges (Fig. 2b). Possibly, they may be required only to form a transient and loose contact between donor and acceptor, prior to arranging into the final conformation, which is competent for electron transfer. A similar conclusion was drawn in a previous NMR study on plant Pc as well[49]. The unexpected cross-link formation by the E69Q/E70Q Cyt $c_6$ variant also suggests different complex assemblies that require further conformational changes, allowing efficient electron transfer (Supplementary Fig. 10).

Our structural data further provide the direct visualization of Cyt $c_6$-R66-mediated interactions, whose functional significance was previously implicated in cyanobacteria (Fig. 2d, Supplementary Fig. 8b)[27,28]. This residue makes a putative electrostatic contact with PsaB-D623, which has already been proposed but not structurally confirmed[13]. At the same time, R66 can engage in a three-layer π(cation)-π interaction along with PsaB-R622 and Cyt $c_6$-W63. The dual interactions mediated by R66 may explain its crucial role. Our mutagenesis analysis indicates that the eukaryotic system also significantly relies on this residue, although its role is less critical compared with its

cyanobacterial counterparts (Fig. 2f, Supplementary Fig. 10)[27,28]. In *Synechocystis* sp. PCC 6803 and *Anabaena* sp. PCC 7119, substitution of the analogous Arg residues diminishes electron transfer rates by ~10 times. Notably, cyanobacterial Pc also contains the corresponding crucial Arg residue (R88), along with the subjacent aromatic residue (Y83), functionally equivalent to Cyt $c_6$-W63 (Supplementary Fig. 1)[44]. This suggests that the cyanobacterial Pc:PSI assembly also utilizes similar three-layer π(cation)-π interactions. Furthermore, both the aromatic residues, Cyt $c_6$-W63 and Pc-Y83, are immediately adjacent to the cofactors, heme and copper, respectively; in other words, the π interactions take place in the vicinity of the redox cofactors (Fig. 2c, d)[44]. This indicates that the π(cation)-π interactions can be modified by electron transfer reactions, which is also supported by the fact that Pc-R88 adopts different conformations in oxidized and reduced Pc[50]. The altered π interactions could modulate the donor:acceptor assembly, and here, it is hypothesized that the π electrons adjust donor:acceptor binding/unbinding according to electron transfer reactions. This could explain how reduced donors efficiently bind to acceptors, while their oxidized form unbinds swiftly, optimizing the turnover of these reactions[6,51]. Interestingly, a similar series of π(cation)-π interactions within the vicinity of electron transfer pathways is also implicated in PSI:ferredoxin interface, hinting at the prevalence of the π-electron-based binding/unbinding switching mechanism[24]. However, it appears that in Chlamydomonas Cyt $c_6$, the functional significance of the π interactions is partly buried by the PSAF-mediated interactions (Fig. 2f, Supplementary Fig. 10). Furthermore, in eukaryotic Pc, the R88 is completely lost and replaced by the long-range electrostatic interactions, which increases binding propensity; it is attributed to the stable donor:PSI complex formation observed in eukaryotes (Supplementary Fig. 1)[20,52].

Together, these findings illuminate an evolutionary trajectory wherein Cyt $c_6$ is gradually replaced by Pc, simultaneously increasing binding affinity and electron transfer efficiency with the emergence of PSAF-dependent interactions. The enhancement of reaction efficiency reduces the required concentration of electron donors, endowing eukaryotes with distinct photosynthetic strategies that economize iron and copper relative to cyanobacteria.

## Methods

### PSI purification

Experiments were performed using the previously mentioned Chlamydomonas strain that expresses PsaB fused with His$_{20}$-Tag after the third residue from the N-terminus[23]. They were kept on Tris-acetate-phosphate (TAP) medium, solidified with 1.5% w/v agar at 25 °C under light of ~20 μmol photons m$^{-2}$ s$^{-1}$. For experiments, the algae were cultured in liquid TAP medium on a rotary shaker (120 rpm) at 25 °C under continuous light of ~20 μmol photons m$^{-2}$ s$^{-1}$.

The PSI purification was performed according to the protocol mentioned previously, with some modifications[23]. The Chlamydomonas cells were harvested under oxic conditions by centrifugation at 1300 $g$ at 4 °C for 5 min (Beckman Colter JLA-16.250 rotor). They were resuspended in ice-cold buffer (25 mM HEPES/KOH, pH 7.5, 0.33 M sucrose, 5 mM MgCl$_2$, 1 mM Phenylmethylsulfonyl fluoride (PMSF), 1 mM benzamidine and 5 mM aminocaproic acid) and homogenized using a nebulizer (2 bar, two passages). The nebulized cells were collected by centrifugation at 48,000 $g$ at 4 °C for 10 min (Beckman Colter JA-25.50 rotor), and the pellets were thoroughly resuspended using a Potter homogenizer in 5 mM HEPES/KOH, pH 7.5, 0.5 M sucrose, 10 mM EDTA, 1 mM benzamidine and 5 mM aminocaproic acid. The resuspended material was layered on top of a sucrose density gradient (1.8 M and 1.3 M sucrose, the same composition as the resuspension buffer). After ultracentrifugation at 98,000 $g$ at 4 °C for 1 h (Beckman Colter SW 32 Ti rotor), the thylakoid membranes were recovered by pipetting the step gradient interphases and

diluted ~4 times, followed by pelleting by centrifugation at 55,000 $g$ at 4 °C for 20 min.

The isolated thylakoids were carefully resuspended using a paint brush and set to 1 mg Chl mL$^{-1}$ in 5 mM HEPES/KOH, pH 7.5, 1 mM benzamidine and 5 mM aminocaproic acid. The chlorophyll concentration was determined using spectrophotometer[53]. The same volume of 2% [w/v] n-Dodecyl α-maltoside (α-DDM) in a buffer of the same composition was added, and the thylakoids were solubilized on ice for 10 min with occasional gentle mixing. Un-solubilized materials were precipitated by centrifuging at 14,000 $g$ at 4 °C for 5 min, and the supernatant was diluted 5 times in 25 mM HEPES/KOH, pH 7.5, 100 mM NaCl, 5 mM MgSO$_4$, 10% [v/v] glycerol, 1 mM benzamidine and 5 mM aminocaproic acid. The sample was loaded onto a TALON metal affinity column (1 ml resin mg Chl$^{-1}$) at a rate of ~0.5 ml min$^{-1}$. The column was washed with ten times the column volume of 25 mM HEPES/KOH, pH 7.5, 100 mM NaCl, 5 mM MgSO$_4$, 10% [v/v] glycerol and 0.02% [w/v] α-DDM at a rate of ~0.5 ml min$^{-1}$ and then washed with the same volume of 25 mM HEPES/KOH, pH 7.5, 100 mM NaCl, 5 mM MgSO$_4$, 10% [v/v] glycerol, 0.02% [w/v] α-DDM and 5 mM imidazole at a rate of ~1 ml min$^{-1}$. The PSI was eluted with 25 mM HEPES/KOH, pH 7.5, 100 mM NaCl, 5 mM MgSO$_4$, 10% [v/v] glycerol, 0.02% [w/v] α-DDM and 150 mM imidazole. The PSI was concentrated with a spin column (regenerated cellulose: 100,000 molecular weight cut-off (MWCO)). The sample was diluted ~10 times with 30 mM HEPES/KOH, pH 7.5, 0.02% [w/v] α-DDM and reconcentrated twice.

### Cyt $c_6$ mutation and purification

To heterologously express Chlamydomonas Cyt $c_6$, the coding DNA sequence (CDS) of the wild type (WT) Cyt $c_6$ was synthesized (GenScript) and ligated into the NdeI/EcoRI restriction sites of the expression vector pET22b. The resulting plasmid was used to transform NEB5α competent *E. coli* cells. Site-directed mutagenesis for different Cyt $c_6$ variants was performed employing In-Fusion Snap Assembly Starter Bundle (Takara). The pET22b-Cyt $c_6$ (WT) was amplified using specific pairs of primers, synthesized by Metabion, carrying the codons for each Cyt $c_6$ variant (Supplementary Table 2). The linear amplicons were assembled into circular plasmids following the manufacturer's instructions, to produce pET22b-Cyt $c_6$ (R66Q), pET22b-Cyt $c_6$ (E69Q), pET22b-Cyt $c_6$ (E70Q), pET22b-Cyt $c_6$ (E71Q) and pET22b-Cyt $c_6$ (E69Q/E70Q). They were used to transform Steller Competent Cells (Takara). The plasmids were isolated from the competent cells and confirmed by sequencing (Eurofins Genomics). For recombinant protein expression, BL21(DE2) competent *E. coli* cells were co-transformed with each pET22b-Cyt $c_6$ and pEC86, which expresses *E. coli* cytochrome c maturation (Ccm) genes[54]. The protein expression was induced by culturing the *E. coli* strains in LB medium containing 1 mM Isopropyl β-D-1-thiogalactopyranoside (IPTG), 50 μgr mL$^{-1}$ Ampicirin and 34 μgr mL$^{-1}$ Chloramphenicol on a rotary shaker (140 rpm) at 30 °C overnight. The cells were harvested by centrifugation at 3700 $g$ at 4 °C for 5 min (Beckman Colter JLA-16.250 rotor). The pelleted cells were resuspended in lysis buffer (20 mM Tricine, pH 7.8, 10 mM KCl and 1 mM PMSF) and sonicated for a total of 180 s (Brason 250 Digital Sonifier w/ Prob, Marshall Scientific). The homogenate was centrifuged at 12,000 $g$ at 4 °C for 30 min (Beckman Colter JA-25.50 rotor), and the supernatant was loaded on an anion exchange column (DEAE Sepharose CL-6B, GE Healthcare) at a rate of ~1 ml min$^{-1}$. The column was washed with ten times the volume of the lysis buffer, and the second wash was done with 20 mM Tricine, pH 7.8 and 50 mM KCl. The Cyt $c_6$ was eluted with 20 mM Tricine, pH 7.8 and 400 mM KCl and was concentrated using a spin column (regenerated cellulose: 3,000 MWCO). The sample was diluted ~10 times with 20 mM Tricine, pH 7.8 and 10 mM KCl and reconcentrated. The sample was further purified employing size exclusion chromatography (Superdex 65 10/300 GL on an AKTA pure system). The concentration of Cyt $c_6$ was determined spectroscopically (at 552 nm with an extinction coefficient of 20 mM$^{-1}$ cm$^{-1}$)[55].

## Cross-linking and Cryo-EM sample preparation

Cyt $c_6$:PSI cross-linking was performed as previously described with some modifications (Supplementary Fig. 2)[23]. The Chlamydomonas WT Cyt $c_6$ was loaded on a PD G25 desalting column and eluted with 10 mM MOPS, pH 6.5, followed by concentration using a spin column (regenerated cellulose: 10,000 MWCO). The Cyt $c_6$ (100 μM) was preactivated by incubating with 5 mM EDC and 10 mM sulfo-NHS for 20 min at room temperature in the dark. Un-reacted crosslinkers were removed using a PD G25 desalting column, and the activated Cyt $c_6$ was concentrated using a spin column (regenerated cellulose: 10,000 MWCO). The activated Cyt $c_6$ (20 μM) and the purified PSI (0.1 mg Chl mL$^{-1}$) were mixed in 30 mM HEPES/KOH, pH 7.5 containing 1 mM ascorbate, 0.1 mM DAD, 3 mM $MgCl_2$ and 0.03% [w/v] α-DDM, and the mixture was incubated for 45 min at room temperature in dark for a cross-linking reaction, followed by purification via a 1.3 M to 0.1 M sucrose density gradient including 5 mM HEPES/KOH, pH 7.5 and 0.02% [w/v] α-DDM ( ~ 60 mg Chl per gradient). The PSI fractions were collected after ultracentrifugation at 180,000 $g$ at 4 °C for 14 h (Beckman Colter SW 41 Ti rotor). For Cryo-EM analysis, the sample was subjected to a PD G25 desalting column to remove sucrose and subsequently concentrated to ~1.6 mg Chl mL$^{-1}$ in 34 μl.

For the crosslinking experiments in Supplementary Fig. 10, non-activated Cyt $c_6$ (4 μM) and PSI (40 μg Chl mL$^{-1}$) were incubated in 30 mM HEPES/KOH, pH 7.5, containing 1 mM ascorbate, 0.1 mM DAD, 3 mM $MgCl_2$, 0.03% [w/v] α-DDM, 5 mM EDC and 10 mM sulfo-NHS for 45 min at room temperature in dark. The cross-linking reaction was quenched with 50 mM Tris, pH 8.0, and the reaction mixture was subjected to SDS-PAGE and Western blotting.

## SDS-PAGE and Western blotting

The protein samples were supplemented with one tenth volume of loading buffer (250 mM Tris/HCl, pH 6.8, 8% [w/v] SDS, 40% [v/v] glycerol and 0.33% [w/v]) and 100 mM DTT and subsequently heated at 65 °C for 20 min. After being separated in SDS-PAGE via electrophoresis, they were stained with Coomassie Serva Blue G or electro-transferred onto nitrocellulose membranes (Amersham). For Western blotting, the membranes were incubated with anti-Cyt $c_6$ antibody (1:3000), which was kindly provided by Sabeeha S.Merchant (University of California). The protein-antibody complexes were labeled with a Supersignal West Pico Plus chemiluminescent substrate (Thermo Scientific).

## Cryo-EM data acquisition, processing and model building

For cryo-EM sample preparation, 4 μl of purified Cyt $c_6$:PSI complex was applied to glow-discharged Quantifoil R 2/1 copper 300-mesh holey carbon grids (PELCO easiGlow, 25 s at 15 mA). Grids were vitrified in liquid ethane-propane using a Vitrobot Mark IV (Thermo Fisher Scientific) at 100% humidity and 4 °C, with a blotting time of 9 s and blotting force 7. Cryo-EM data were collected on a Titan Krios G4 transmission electron microscope (Thermo Fisher Scientific) operated at 300 kV, equipped with a Selectris energy filter and Falcon 4 direct electron detector. Movies were acquired in counting mode at a nominal magnification of 165,000×, yielding a calibrated pixel size of 0.73 Å. Each exposure was 1.9 s and split into 65 frames, with a total accumulated dose of 50 e–/Å$^2$. In total, 21,413 movies were collected.

All cryo-EM data processing was performed in CryoSPARC[35]. Micrographs were gain-corrected, aligned, and dose-weighted using Patch Motion Correction[56]. Thereafter, the contrast transfer function (CTF) was estimated using Patch CTF[57]. After discarding micrographs with poor CTF fits or suboptimal quality, 21,147 micrographs were retained for particle picking. Initial particle picking was carried out on a subset of 1000 micrographs using the Blob picker. 2D classes generated from this subset were used

to train a Topaz model[58,59], which underwent four rounds of iterative training. The final Topaz model was used to extract particles on the entire dataset, resulting in approximately 1.3 million particles. Particles were extracted with a 512-pixel box size, binned 4 times for initial classification. Extracted particles were subjected to 2D classification to remove contaminants. The retained particles after the 2D classifications were then subjected to five ab initio 3D reconstructions, resulting in both high-quality and junk classes. These reconstructions were refined through heterogeneous refinement. Multiple rounds of ab initio heterogeneous refinement were performed to systematically remove poor-quality classes. After these steps, around 246,000 high-quality particles remained and were re-extracted at full (unbinned) 512-pixel box size for high-resolution refinement. These particles were refined by non-uniform refinement in CryoSPARC, yielding a reconstruction at 1.83 Å global resolution (PSI map), as estimated with the gold-standard Fourier shell correlation (FSC) with a cut-off of 0.143. At this point, the Cyt $c_6$ component was not well resolved, a mask was generated around Cyt $c_6$, followed by masked 3D classification into six classes. In one class, Cyt $c_6$ showed a clear well-resolved density; this class was further refined locally using the focused mask, giving a 2.06 Å map (Cyt $c_6$-PSI-core map) where side chains, pigments, cofactors, and the crosslink between PsaF and Cyt $c_6$ were well resolved (Supplementary Fig. 3). Two additional densities were observed on the stromal side of the complex. To further resolve these densities, focused classification was performed; however, due to extreme flexibility, they could not be refined. The resulting PSI and Cyt $c_6$-PSI-core maps were used for model building.

For the PSI map, the previously determined Chlamydomonas PSI structure (PDB ID: 7ZQC) was used as the starting model[23]. For the Cyt $c_6$-PSI-core map, the same PSI structure was combined with the crystal structure of Cyt $c_6$ (PDB ID: 1CYI)[37]. Both models were rigid-body fitted into the 1.83 Å and 2.06 Å maps, respectively, using UCSF ChimeraX v1.10[60]. Side chains and ligands were inspected and fitted manually in Coot v0.9.8.9[61,62]. Given the high resolution, water molecules were modeled automatically using the water picking tool in Coot, and each placement was inspected manually. Final models were refined using real-space refinement in PHENIX v1.21.1-4907[63]. CIF restraints for chlorophyll B (CHL) and chlorophyll A isomer (CL0) were generated with the Grade Web Server (https://grade.globalphasing.org/), while those for lutein (LUT) and neoxanthin (NEX) were created using the eLBOW utility in PHENIX by providing the chemical file (mol2) and final geometry file from the PDB models as inputs. Before PDB submission, multiple rounds of validation and model building were carried out using Phenix and Coot. And finally, all figures were prepared using UCSF ChimeraX.

## Single-flash absorbance spectroscopy

Kinetic studies of P700 post laser flash re-reduction were conducted as described previously[39]. In short, isolated PSI complexes (331 nM) were solubilized in 7.5 mM KCl, 2.5 mM $MgCl_2$, 10 mM ascorbate, 0.5 mM DAD and 1 mM methyl viologen, pH 7.0 (MOPS 5 mM). The mixture was placed in a "Joliot type Spectrophotometer" (JTS-150, Biologics) supplied with a "Smart-Lamp" with a dual measuring light usage (705 to 740 nm) and adequate detector filters (P700). Absorbance was measured post a laser flash (100 detections in a decreased exponential rate for the duration of 5 s with an initial delay of 700 μs, 4 technical repetitions per test). Cyt $c_6$ was gradually added (0.5–4.0 μM), and $k_2$ values were calculated using Origin. Pro (ExpDec1). For generating the boxplots, the $k_2$ values for 2–4 μM Cyt $c_6$ were averaged, using 3 biological replicates of independent PSI isolations. For statistical comparisons, a One-way ANOVA with Dunnett multiple comparisons test was applied using Biorender.com.

## Reporting summary

Further information on research design is available in the Nature Portfolio Reporting Summary linked to this article.

## Data availability

Structural data produced in this work, including cryo-EM density map and refined atomic coordinates, have been deposited in the Electron Microscopy Data Bank and the Protein Data Bank under the accession codes: 9SE6 and EMDB-54803 [https://www.ebi.ac.uk/pdbe/entry/emdb/EMD-54803] (PSI) and 9SE7 and EMDB-54804 [https://www.ebi.ac.uk/pdbe/entry/emdb/EMD-54804] (PSI-Cyt $c_6$). Source data are provided with this paper.

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

## Acknowledgments

We thank Christoph Gerle for constructive and insightful discussions, Laura Mosebach and Vida Adlfar for experimental supervision, Samuel Wink for facilitating Cytochrome $c_6$ expression and Rémi Ruedas and Mohamed Chami for collecting the cryo-EM data at the BioEM lab of the Biozentrum, University of Basel. We also thank Anuj Kumar for valuable suggestions during cryo-EM data processing and model building. This work was financially supported by FOR 5573/1 GoPMF (DFG HI 739/25.1) and DFG HI 739/13.3 to M.H., and by FOR 5573/1 GoPMF (DFG SCHU 3364/3-1) and LOEWE RobuCop to J.M.S.

## Author contributions

M.H. and J.M.S. designed the project; G.K. provided the Cytochrome $c_6$ expression system; Y.O. prepared the sample for cryo-EM analysis; G.P.M. prepared the cryo-EM grids, collected and processed the cryo-EM data and built the model; Y.O., Y.M., and M.S. performed biochemical analysis; Y.O., G.P.M., Y.M., M.H., and J.M.S. interpreted the data and wrote the manuscript. All authors contributed to the analysis and the final version of the manuscript.

## Funding

## Competing interests

The authors declare no competing interests.
