## [Transparent Peer Review file · Nature Communications]

Cryo-EM structure of *Chlamydomonas reinhardtii* Photosystem I complexed with cytochrome c6

Corresponding Author: Dr Jan Schuller

Version 0:

Reviewer comments:

Reviewer #1

(Remarks to the Author)

This manuscript describes a structural study of PSI in complex with its electron donor Cyt c6 by cryo-EM. The PSI-Cyt c6 complex was obtained by cross-linking, and the structure was solved at a high resolution, allowing the authors to reveal the structural details of the interaction of Cyt c6 with PSI. The results are largely consistent with previous functional studies on the electron transfer between Cyt c6 and PSI in green algae. The authors also compared the features of green algal Cyt c6 with its cyanobacterial counterpart, and suggested the similarities and differences between these two organisms.

Overall, the results obtained are new and important, and are therefore worth publishing. However, I would like to ask the authors to address the following points and revise their manuscript accordingly.

There should be an Extended Data Table (Table 1) which summarizes the cryo-EM data collection and structure analysis statistics. Please supply such a Table. The structure was said to have a resolution of 1.83 Å, but judging from the cryo-EM map quality (for example, the map of the chlorophylls) shown in Extended Data Fig. 4, it may not have such a high quality, as the cryo-EM map for chlorophylls usually is empty in the middle of the chlorin rings at this resolution.

Judging from the Fig. 1B, the amount of the crosslinked PsaF seems to be equivalent or less than the free PsaF, which means that the amount of PSI without Cyt c6 should be equivalent or more than that of PSI with Cyt c6. Did the authors observe PSI particles without Cyt c6 separately in their SPA analysis?

Line 63 and throughout the text: What is "PSAF"? If it is a product of the gene *psaf*, it should be written as "PsaF". Also for "PSAD" in line 113.

Line 87: Please spell out "EDC".

Line 92: "...with the major PSI fraction recovered at lower density (Fig. 1a)". Where is the band for the original PSI? If you don't know the band position of the original PSI, how could you know that the PSI-Cyt c6 band is lower than the original PSI?

Lines 108-109: "This yielded a locally enhanced reconstruction at 2.06 Å, in which an additional density adjacent to PSAF was clearly resolved (Extended Data Fig. 3)". In Extended Data Fig. 3, the Cyt c6 part seems to have been enlarged, but it is not clear that this part corresponds to which area in the PSI complex shown in Extended Data Fig. 3A. The area corresponding to Extended Data Fig. 3B should be indicated in Extended Data Fig. 3A, or be shown separately.

Line 152: "...PsaB-Q604/606". What is the residue "606"?

Line 219: "on" should be "an". Line 220: "greater", better to use "larger".

Line 222-224: "This asymmetry may explain the stronger functional...". The logic of this sentence is not clear to this reviewer, as it was described that the interaction of Cyt c6 is stronger with PsaB than with PsaA, why mutation of PsaA-W651 would have a stronger impact than mutation of PsaB-W626?

Line 315: Please spell out "DDM".

Line 333: What is "CDS

Line 346: What are "ccm" genes?

Line 442: "fitted".

Line 455: Change "we" to "were".

Extended Data Fig. 5, legend: "Gray and black brackets indicate PsaA-I loop and 551 PsaB-I' loop, respectively." Where is "gray and black brackets", and what are the color means?

Reviewer #2

(Remarks to the Author)

Through biochemistry and structural biology, this manuscript aims to reveal how cyt c6 binds to PSI in the model green alga *Chlamydomonas*. As the authors nicely explain, different photosynthetic organisms have evolved alternative PSI electron donors. However, plants can only use plastocyanin. Understanding this transition is of significant interest to the field.

Since the cyt c6/PSI interaction is transient, the authors turned to chemical cross-linking. Exposed carboxylates on recombinant cyt c6 were activated with EDC and NHS. When mixed with purified *Chlamydomonas* PSI, the activated cyt c6 carboxylates could react with exposed PSI primary amines.

One such connection was observed between PsaF-K26 and cyt c6-E69. Other noncovalent interactions were observed in the surrounding region including an interesting effect involving cyt c6-R66 and both negative and aromatic groups on PSI.

I have one major conceptual concern with this work. In my understanding, the exposed cyt c6 carboxylates are neutralized upon EDC/NHS treatment and exist as transient NHS esters. However, the authors discuss at length the negative charges of cyt c6 Glu residues and how they interact with PSI. In the figures, E69-71 are shown as carboxylates. Is this correct? Are the long distances shown in Figure 2B a result of this modification? I understand that these NHS esters are transient, but their existence is essential to the methodology used. I challenge the authors to add sufficient text to the results and discussion to address this concern and so that future readers will not share my confusion. It is essential that they reevaluate the local electron density maps to see if any hints of NHS esters are observed.

As minor points:

- (1) Please use consistent PSI protein subunit nomenclature (e.g. PsaF not PSAF);
- (2) Say more about the interesting result involving the E69Q/E70Q double mutant in vitro result. What possible compensatory interactions might be occurring?
- (3) Check usage, spelling, and appropriate abbreviations in the cyt 6 mutation and purification section of methods.
- (4) In general, figure captions need to be more descriptive. For example, Figure 2's caption doesn't fully explain the colors that represent PSI and cyt c6.
- (5) Methods about Western blotting are quite inadequate. Which antibodies were used?!

Version 1:

Reviewer comments:

Reviewer #1

(Remarks to the Author)

The authors have addressed my concerns adequately, and I have no further questions.

Reviewer #2

(Remarks to the Author)

The authors have thoroughly addressed all points I previously raised.

REVIEWERS COMMENTS

Reviewer #1:

General comments

This manuscript describes a structural study of PSI in complex with its electron donor Cyt c₆ by cryo-EM. The PSI-Cyt c₆ complex was obtained by cross-linking, and the structure was solved at a high resolution, allowing the authors to reveal the structural details of the interaction of Cyt c₆ with PSI. The results are largely consistent with previous functional studies on the electron transfer between Cyt c₆ and PSI in green algae. The authors also compared the features of green algal Cyt c₆ with its cyanobacterial counterpart and suggested the similarities and differences between these two organisms.

Major concerns

Overall, the results obtained are new and important and are therefore worth publishing. However, I would like to ask the authors to address the following points and revise their manuscript accordingly.

- There should be an Extended Data Table (Table 1) which summarizes the cryo-EM data collection and structure analysis statistics. Please supply such a Table.

Thank you for pointing this out. We had prepared the table but inadvertently omitted it from the submission - our apologies for the oversight. We have now included Supplementary Table 1: Cryo-EM data collection, refinement, and validation statistics with the revised manuscript.

- The structure was said to have a resolution of 1.83 Å, but judging from the cryo-EM map quality (for example, the map of the chlorophylls) shown in Extended Data Fig. 4, it may not have such a high quality, as the cryo-EM map for chlorophylls usually is empty in the middle of the chlorin rings at this resolution.

Thank you for this insightful comment. You are correct that, at this resolution, the cryo-EM density for chlorophylls typically shows an empty center within the chlorin rings. In the originally submitted figures, we used the non-sharpened maps, which can obscure this feature. We have updated Extended Data Fig. 4 (now Supplementary Figure 5) using the sharpened maps, and the characteristic empty density in the centers of the chlorin rings is clearly visible.

- Judging from the Fig. 1B, the amount of the crosslinked PsaF seems to be equivalent or less than the free PsaF, which means that the amount of PSI without Cyt c₆ should be equivalent or more than that of PSI with Cyt c₆. Did the authors observe PSI particles without Cyt c₆ separately in their SPA analysis?

Thank you for this thoughtful observation. Indeed, based on the crosslinking results, a fraction of PSI particles in the dataset lacked bound Cyt c₆. During SPA processing, we identified this particle population and performed a separate reconstruction. This reconstruction revealed no structural differences compared with the Cyt c₆-bound PSI complex.

Minor issues

- Line 63 and throughout the text: What is “PSAF”? If it is a product of the gene *psaf*, it should be written as “PsaF”. Also for “PSAD” in line 113.

PSAF and PSAD are nuclear-encoded and therefore written in uppercase, whereas chloroplast-encoded genes such as *psaB* are written as PsaB.

- Line 87: Please spell out “EDC”.

Done.

- Line 92: “...with the major PSI fraction recovered at lower density (Fig. 1a).”. Where is the band for the original PSI? If you don’t know the band position of the original PSI, how could you know that the PSI-Cyt *b6* band is lower than the original PSI?

Here, we confined it to “The cross-linked Cyt *c6*:PSI complex was subjected to sucrose density gradient (SDG) centrifugation to separate it from the non-cross-linked Cyt *c6* (Fig. 1a).” However, we found that PSI fraction migrated to lower densities compared with our previous experiments (Naschberger et al., Nat. Plants, 2022), and it is mentioned in the subsequent paragraph referring to stromal densities.

- Lines 108-109: “This yielded a locally enhanced reconstruction at 2.06 Å, in which an additional density adjacent to PSAF was clearly resolved (Extended Data Fig. 3).”. In Extended Data Fig. 3, the Cyt *c6* part seems to have been enlarged, but it is not clear that this part corresponds to which area in the PSI complex shown in Extended Data Fig. 3A.

Thank you for pointing this out. We agree that the relationship between the zoomed-in view and its location on the full PSI complex was not sufficiently clear. We have now updated Extended Data Fig. 3 (now Supplementary Figure 4) to include a highlighted inset and arrows indicating the precise region adjacent to PsaF from which the enlarged Cyt *c6* density was taken. This should clarify the spatial context of the locally enhanced reconstruction.

- The area corresponding to Extended Data Fig. 3B should be indicated in Extended Data Fig. 3A, or be shown separately.

We modified the Extended Data Fig. 3 accordingly (see Supplementary Fig. 4)

- Line 152: “...PsaB-Q604/606”. What is the residue “606”?

This was revised to “N606”, it is also shown in Supplementary Fig. 8b and 9c (middle).

- Line 219: “on” should be “an”.

Done

- Line 220: “greater”, better to use “larger”.

Done

- Line 222-224: “This asymmetry may explain the stronger functional...”. The logic of this sentence is not clear to this reviewer, as it was described that the interaction of Cyt *c6* is stronger with PsaB than with PsaA, why mutation of PsaA-W651 would have a stronger impact than mutation of PsaB-W626?

This part was revised “Relying on fewer contact sites in interacting with PsaA may explain the stronger functional impacts of mutations on PsaA-W651 relative to PsaB-W626.”

- Line 315: Please spell out “DDM”.

Spelled out

- Line 333: What is “CDS
coding DNA sequence
- Line 346: What are “ccm” genes?
cytochrome c maturation
- Line 442: “fitted”.
Thanks, corrected
- Line 455: Change “we” to “were”.
Thanks, corrected
- Extended Data Fig. 5, legend: “Gray and black brackets indicate PsaA-I loop and 551 PsaB-I’ loop, respectively.” Where is “gray and black brackets”, and what are the color means?
We have modified the figure (see Supplementary Fig. 7b). In the new figure, we marked the I/I’ loops in corresponding-coloured squares. This was done to highlight the surrounding region of the tryptophan dimer.

Reviewer #2:

General comments

Through biochemistry and structural biology, this manuscript aims to reveal how cyt c6 binds to PSI in the model green alga *Chlamydomonas*. As the authors nicely explain, different photosynthetic organisms have evolved alternative PSI electron donors. However, plants can only use plastocyanin. Understanding this transition is of significant interest to the field. Since the cyt c6/PSI interaction is transient, the authors turned to chemical cross linking. Exposed carboxylates on recombinant cyt c6 were activated with EDC and NHS. When mixed with purified *Chlamydomonas* PSI, the activated cyt c6 carboxylates could react with exposed PSI primary amines. One such connection was observed between PsaF-K26 and cyt c6-E69. Other noncovalent interactions were observed in the surrounding region including an interesting effect involving cyt c6-R66 and both negative and aromatic groups on PSI.

Major concerns

I have one major conceptual concern with this work. In my understanding, the exposed cyt c6 carboxylates are neutralized upon EDC/NHS treatment and exist as transient NHS esters. However, the authors discuss at length the negative charges of cyt c6 Glu residues and how they interact with PSI. In the figures, E69-71 are shown as carboxylates. Is this correct? Are the long distances shown in Figure 2B a result of this modification? I understand that these NHS esters are transient, but their existence is essential to the methodology used. I challenge the authors to add sufficient text to the results and discussion to address this concern and so that future readers will not share my confusion. It is essential that they reevaluate the local electron density maps to see if any hints of NHS esters are observed.

Thank you very much for your comment. We added a new figure “Supplementary Fig. 2” and explained the crosslinking procedure in more detail as follows “To obtain structural insights into the Cyt *c*₆:PSI interactions, we reconstituted a *Chlamydomonas* Cyt *c*₆:PSI complex by chemical crosslinking (Supplementary Fig. 2). PSI (PsaB-His₂₀) was affinity-purified from *Chlamydomonas* cells grown under standard conditions, while *Chlamydomonas* Cyt *c*₆ was recombinantly expressed, purified, and pre-activated with 1-ethyl-3-[3-dimethylaminopropyl] carbodiimide (EDC) and *N*-hydroxysulfosuccinimide (sulfo-NHS)²³.

This converts the Cyt *c*₆ carboxyl groups into a short-lived sulfo-NHS ester intermediate (Supplementary Fig. 2a). After removing excess crosslinking agents, the activated Cyt *c*₆ was incubated with isolated PSI complex (Supplementary Fig. 2b,c). As Cyt *c*₆:PSI complexes form, sulfo-NHS esters located close to PSI amino groups can react to produce amide bonds, enabling zero-length crosslinking, while the esters that are not in close vicinity are rapidly hydrolyzed without forming crosslinks (Supplementary Fig. 2c)^{17,18,26,29,30}. This stabilizes the donor:PSI complexes at the positions where carboxyl and amino groups are in close proximity. Importantly, this type of cross-linking preserves the native Cyt *c*₆:PSI complex conformation that is competent in rapid electron transfer^{18,26}."

Minor issues

- Please use consistent PSI protein subunit nomenclature (e.g. PsaF not PSAF);
PSAF and PSAD are nuclear-encoded and therefore written in uppercase, whereas chloroplast-encoded genes such as *psaB* are written as PsaB.
- Say more about the interesting result involving the E69Q/E70Q double mutant in vitro result. What possible compensatory interactions might be occurring?

We revised as follows "Interestingly, the phenotype of E69Q/E70Q double mutant was weaker than those of each single mutant, suggesting formation of donor:acceptor complexes in different conformational states which are less competent for the electron transfer reactions." We did not mention in the text, but the lack of E69/E70 charged might force their partners (PSAF-K23/27) to interact with other negatively charged residues (like Cyt *c*₆-E71), resulting in donor:acceptor assembly in non-productive conformations.

- Check usage, spelling, and appropriate abbreviations in the cyt 6 mutation and purification section of methods.

Done

- In general, figure captions need to be more descriptive. For example, Figure 2's caption doesn't fully explain the colors that represent PSI and cyt *c*₆.

Thank you for pointing this out. We added detailed explanations.

- Methods about Western blotting are quite inadequate. Which antibodies were used?

Thank you, this information was added. "For Western blotting, the membranes were incubated with anti-Cyt *c*₆ antibody, which was kindly provided by Sabeeha S.Merchant (University of California)."